# Land-use and socioeconomic time-series reveal legacy of redlining on present-day gentrification within a growing United States city

Peter C. Ibsen[1*⊙], Anna Bierbrauer[2,3⊙], Lucila M. Corro[1], Zachary H. Ancona[1], Mark Drummond[1], Kenneth J. Bagstad[1], Jay E. Diffendorfer[1]

1 U.S. Geological Survey, Geosciences & Environmental Change Science Center, Denver, Colorado, United States of America, 2 University of Colorado Denver, College of Architecture and Planning, Denver, Colorado, United States of America, 3 University of Wisconsin Madison, Department of Planning and Architecture, Madison Wisconsin, United States of America

⊙ These authors contributed equally to this work.
* pibsen@usgs.gov

## Abstract

Home Owners' Loan Corporation (HOLC) maps illustrated patterns of segregation in United States cites in the 1930s. As the causes and drivers of demographic and land-use segregation vary over years, these maps provide an important spatial lens in determining how patterns of segregation spatially and temporally developed during the past century. Using a high-resolution land-use time series (1937-2018) of Denver, Colorado, USA, in conjunction with 80 years of U.S. Census data, we found divergent land-use and demographics patterns across HOLC categories were both pre-existent to the establishment of HOLC mapping and continued to develop over time. Over this period, areas deemed "declining" or "hazardous" had more diverse land use compared to "desirable" areas. "Desirable" areas were dominated by one land-use type (single-family residential), while single-family residential diminished in prominence in the "declining/hazardous" areas. This divergence became more established decades after HOLC mapping, with impact to racial metrics and low-income households. We found changes in these demographic patterns also occurred between 2000 and 2019, highlighting how processes like gentrification can develop from both rapid demographic and land-use changes. This study demonstrates how the legacy of urban segregation develops over decades and can simultaneously persist in some neighborhoods while providing openings for fast-paced gentrification in others.

## Introduction

The recent availability of easy-to-access historical Home Owners' Loan Corporation (HOLC) redlining data and HOLC-designated boundaries from the "Mapping Inequality" project [1] has resulted in new examinations of health, spatial, and ecological outcomes associated with the historical housing segregation mapped out by HOLC areas [2–4]. Present day land-use

**Data availability statement:** All relevant data are within the paper and its Supporting Information files and are also available on the website https://www.sciencebase.gov/catalog/item/6712935ed34eb6a152fc6d2a or doi:10.5066/P1RJMRQ5

**Funding:** The author(s) received no specific funding for this work.

**Competing interests:** The authors have declared that no competing interests exist.

and land-cover characteristics such as lower tree canopy coverage, amount of green space and parks, and their associated concerns such as urban heat islands, poor air quality, and environmental pollution have all been connected to HOLC spatial extents [5,6]. However, some of this recent work has been critiqued as being either [1] too casual (i.e., HOLC maps have caused present day pattern) in their interpretations [7], or [2] reliant on limited analyses including poor consideration of patterns found in the different HOLC categories, minimal temporal analysis, and omitting narrative data found in HOLC records' descriptions of neighborhoods [8]. Therefore, to better understand why current economical, physical, and ecological inequities exist, researchers could assess how patterns of inequality developed over time.

Until now, studying urban land-use and land-cover patterns over time has been difficult due to the lack of high-resolution historical land-use data spanning multiple decades. In this paper, we quantify the spatial and socioeconomic patterns from approximately when HOLC maps were created through the following decades using a novel U.S. Geological Survey (USGS) dataset tracking land use in the Denver metropolitan region from 1937 to 1997 [9], coupled with historical U.S. Census data obtained from the IPUMS (formally known as the Integrated Public Use Microdata Series) National Historic Geographic Information System at the University of Minnesota (www.nhgis.org) [10]. Relating multi-decadal land-use data, demographic patterns, and HOLC-designated boundaries identifies linkages between patterns developed during the periods that HOLC influenced lending practices and subsequent trajectories of investment and disinvestment within former HOLC-graded neighborhoods.

Multiple federal and local policies contribute to the patterns of racial and ethnic segregation in U.S. cities before and after the 1930s. 'Redlining' maps—the HOLC color-coded maps rating local mortgage markets on sociodemographic variables such as race, ethnicity, and income as well as housing stock conditions created to allow for mortgage refinancing in the wake of the Great Depression [7]—are regularly studied because they provide clear spatial boundaries in hundreds of cities. However, neighborhood racial segregation existed before the creation of HOLC in the form of racial covenants and other localized discriminatory housing practices [11,12], and there is growing consensus that HOLC policies reinforced, rather than created, inequitable access to affordable mortgage insurance [2,13]. Especially, as described in Hillier 2003, there is little evidence that HOLC maps were actively used in categorically denying mortgages to redlined areas, although discrimination could have still be reinforced at the individual homeowner-scale as local brokers took over foreclosed properties [13]. In the decades following HOLC mapping many other trends and policies influenced patterns of investment and disinvestment in U.S. cities. Urban renewal projects of the 1950s–70s resulted in large physical changes in urban centers, largely in non-White dominated districts [14,15]. From the 1950s to the 1980s urban White residents left city centers for the suburbs, while Black Americans moved from the rural south to cities during a second wave of the "Great Migration" cementing racial segregation within urban areas [16].

Therefore, present-day demographic and land-use spatial patterns in cities reflect past urban policies intended to create long-term changes [14]. When such policies built on existing patterns of segregation or were biased toward particular racial or ethnic groups, neighborhood outcomes diverged [17]. Biased policies and practices have been observed across many nations. In Germany, renters with Turkish last names and pronounced accents were less likely to find rental units [18]. Whereas, in the Netherlands, ethnic minorities applying for loans in "bad" neighborhoods were screened more rigorously [19]. Research in post-apartheid South Africa found that perceptions of a neighborhood's value drove home loan approval rates not the personal and private wealth investments actually present [20].

Land-use and land-cover characteristics associated with inequitable housing policies such as higher multi-family residential density, lower tree canopy density, and higher urban heat

vulnerability [3,12,17,21–23] also make these neighborhoods vulnerable to gentrification [24]. Gentrification and displacement, the processes of higher-income and more-educated residents moving into lower-income neighborhoods, are spurred by variations in housing-stock age, vacancy rates [25], and city and neighborhood-specific factors [26–28]. As reinvestments in U.S. urban centers increase, these types of land-use and land-cover characteristics may be affected differently by changes to zoning codes, speculative real estate development, or re-greening efforts [29–32]. The spatial patterns of this connection between the built environment, past discriminatory policies, and the potential vulnerability to neighborhood gentrification have been connected from the present to the HOLC boundaries [33], but to understand how these spatial patterns emerged over time, time-series data are needed.

Our work addresses the gap in knowledge and data about how urban demographic and land-use patterns developed over time, how they were reified by redlining policies, and reinforced by continual urban renewal policies. Moreover, the diverging demographic and land-use patterns created ideal conditions for extensive changes in present day neighborhood socio-economic demographics through rapid parcel-level reinvestment leading to neighborhood gentrification. Because potentially demographic exclusionary policies still exist, or access to housing is protected through discriminatory gate-keepers in nations worldwide [19,34,35], an understanding of how these patterns occur over time, and the possible outcomes of such practices may only be noticeable long after the fact. Long-term studies can thus provide key information in understanding how urban areas subject to discriminatory housing policies develop and change their demographics and land-use over multi-decadal periods. By using HOLC maps as a starting point, our data analysis can analyze how unique patterns of land use and demographics developed and apply this analysis to anticipate present day patterns of gentrification.

Through our data synthesis, in a case study for Denver, Colorado, we address the knowledge gap of redlining effects through time, rather than examining the present-day conditions. We furthermore examine how redlining legacies influence the often interrelated patterns of land use and urban demographics. In this study, we ask two key research questions regarding the legacy of outcomes resulting from housing segregation and potentially unequal lending programs:

1. How have patterns in land use and demographics developed and diverged in formally categorized HOLC areas across Denver from 1937 to 2019?

2. How have patterns of inequality in land use and demographics from the previous decades influenced rapid present-day changes in Denver's demographics and resulting neighborhood-specific patterns in gentrification?

## Methods

### Study area

Denver is located at the foot of the Rocky Mountains of Colorado, bordered by the mountains to the west and the High Plains to the east. The Denver metropolitan statistical area's annual growth rate of 1.48% makes it one of the fastest growing cities in the western United States, and its current population is 715,000 [36]. Although Denver has a densely developed urban downtown, most of the region is characterized by pockets of medium density surrounded by large tracts of suburban and peri-urban development. Growth across the Denver region has increased steadily since the mid-1850s when White settlers first arrived in the area. Outside the city's boundaries, most land use was agricultural—market farms, large homesteads, and small ranches—and small rural communities. A large influx of federal workers into Denver

before and after World War II boosted demand for single-family housing, and the Federal Housing Administration (FHA) helped defense workers purchase homes, contributing to an already large federal presence [37]. Between 1940 and 1950, Denver's population grew by 29%, and the city itself expanded in area from 58 to 99 square miles [38].

As the Denver metropolitan area expanded and its suburban population grew, land use and demographics changed. Neighborhoods in the periphery of Denver experienced greater upward income mobility compared to the central urban core [39]. In 1958, the Denver Urban Renewal Authority (DURA) was created, with a primary task to "eliminate post World War II slum housing conditions. [40]" This task coincided with the passage of the Federal Housing Act of 1949, which provided billions of dollars to local municipalities to redevelop areas termed "blighted" [41]. Over the following decades, Denver, and other cities across the country, entered a period of "urban renewal" and attempted to increase the property value of its urban core [14]. Into the start of the 21st century, Denver's population and income grew substantially [39], maintaining growth through the Great Recession (9.5% growth between 2000 and 2010), and boomed afterwards (17.7% growth between 2010 and 2020). During the same period, suburban growth was rapid but declined from the 2000s to the 2010s (24.7% between 2000 and 2010 to 15.6% growth from 2010 to 2020) [42].

## Analysis of land-use diversity

To better understand present-day inequities in the United States' cities, we examine a case study of the progressive, long-term impacts of discriminatory policies through land-use time-series data with the historical HOLC-designated boundaries in Denver Colorado. We identify the relative proportions of land-use classes within each HOLC category and track how those land-use patterns and demographics shifted over time (Fig. 1).

The USGS dataset by Drummond et al. [9] derived detailed land-use information for the years 1937, 1957, 1977, and 1997 in Colorado's Front Range. The result is a 1-meter resolution land-use map, with a minimum mapping unit of approximately 2.5 acres, comprising 33 land-use classes. Because the Denver HOLC map was created in 1938, we can understand the pre-existing patterns of land use before any possible influence by HOLC and identify how those patterns matched how the HOLC categories were appraised from each HOLC area's description sheet.

We first determined the area and percentage of land-use classes found in each HOLC category in 1937, 1957, 1977, and 1997. For visual aid to the reader, we have changed the original colors used by HOLC (A = Green, B = Blue, C = Yellow, D = Red), to a color-blind friendly palette, while keeping the D category red (A = Dark Blue, B = Light Blue, C = Orange, D = Red)). We overlaid HOLC maps on the land-use data for each decade and used the Tabulate Area tool in ArcMap 10.8 (Esri Redlands) to determine the proportion of each land-use class within a HOLC area. Both HOLC and land-use data were processed in projected coordinate system NAD 1983 UTM Zone 13N. We examined land-use patterns as the diversity of land use in each HOLC category over time, using metrics of alpha diversity (the Shannon-Wiener index [43]). Although a foundational metric in information theory, the Shannon-Wiener index is also often used in ecological literature to describe biological diversity, and has been applied to studies examining the landscape patterns of urban land use and land cover, both at the city [44,45] and neighborhood scales [46]. In our application, the Shannon-Wiener index (H') described both the proportion of land-use class within an area, and the relative frequency of those land-use classes.

A high H' value indicates more land-use classes evenly distributed within a HOLC category, while a low value indicates relatively few land-use classes, which may be distributed

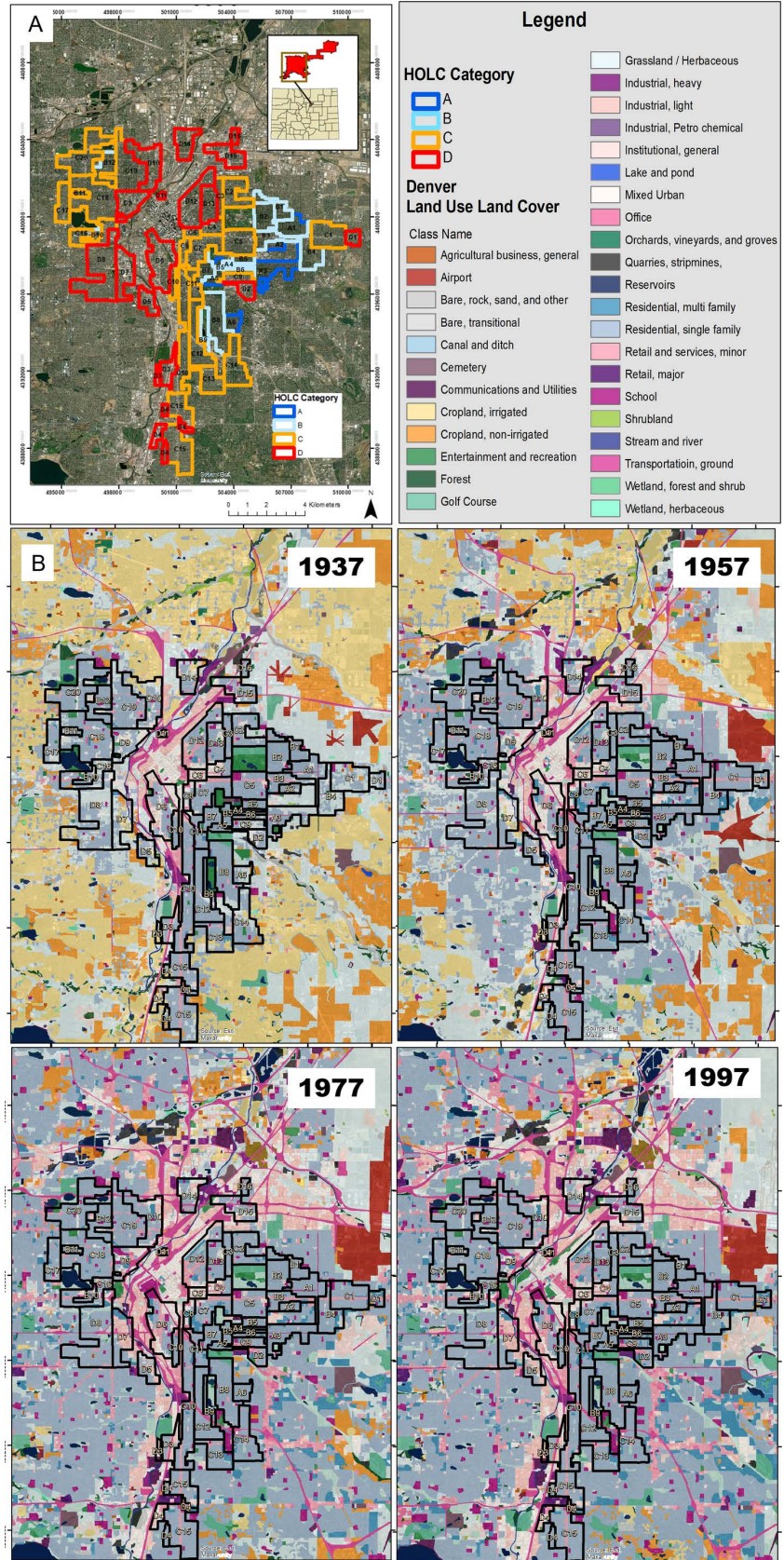

**Fig 1. High-resolution land-use data of Denver from 1937–1997.** A: Location of Home Owners' Loan Corporation (HOLC) categories on a present-day map of Denver, Colorado. B, Land-use data, for each 20-year period (1937, 1957,

1977, 1997), from Drummond et al., 2019 [9], with outlines of each HOLC area and their corresponding identification. The legend included refers to these land-use categories. Basemap from U.S. Geological Survey, The National Map, 2024, HOLC outlines are republished from Mapping Inequality [1] under a CC BY license, with permission from Robert K. Nelson, original copyright 2023.

in uneven groupings. H' is resistant to configurational shifts within an area, as compared to other urban land-use diversity metrics [47], which makes H' more suitable for analyzing the overall change in land-use composition at a neighborhood scale. We tested for statistical significance in the HOLC categories and Year of Analysis influence on Shannon Index through a two-way Analysis of Variance test (ANOVA), and we tested statistical significance between specific HOLC categories using Tukey's Honest Significant Differences (HSD) test, a method regularly used with analyzing post-hoc comparisons among HOLC categories [17,48]. We used a threshold of $p < 0.05$ throughout to determine significance.

To extend the time-series into the current era (~2020), we first added to the Drummond et al. (2019) land-use dataset [9] with a recent land-use dataset from 2018 that used the same resolution (1 m) to allow additional change analyses between 1997 and 2018, a period of rapid urban growth[49]. However, because Drummond et al. [9] and Chen et al. [49] used different classification methodologies and land-use naming conventions, we only included the comparison of residential land use. Secondly, we analyzed the potential influence of zoning, both on current land-use configuration and how zoning changes are distributed across in formally HOLC-designated areas. Zoning data were provided by the City of Denver, and although these data do not directly sync up to our demographic and land-use time-series, they do provide current patterns of zoning designation, as well as the dates at which zoning changes occurred within a parcel dating from 1955–2019. However, the zoning data do not provide a description of what the designation was changed from.

## Analysis of demography time series

We examined demography related to HOLC redlining by deriving data on race, education, and income from the 1940, 1960, 1980, and 2000 U.S. Census through IPUMS [10]. We used these years as the decennial Census events most closely matching years for which we had land-use data. To examine the most current state of demographic patterns in Denver, we used the 2015–2019 American Community Survey [50]. We used weighted tract-level percentage of non-White population as our key racial demographic descriptor, as racial descriptions that specifically identify Black, Indigenous, and other People of Color have changed over the course of the decennial Census and are difficult to compare across decades. For education, we used the tract-level percentage of college-educated individuals over 25 years old. To extract HOLC-level data, we overlaid all tract-level demographics by the HOLC maps and spatially joined to the Census tract level for each decennial Census. Then, using the Tabulate Intersection tool in ArcMap 10.8, we extracted the area of each Census tract contained by a HOLC area. We used this area as a weighting value to calculate the demographic percentages within each HOLC area.

Because the 1940 decennial Census did not include tract-level data on number of households within income brackets, we limited the income analyses to the 1960, 1980, 2000 decennial Census and 2015–2019 American Community Survey. This approach allowed for the comparison of the number of households within income brackets across 59 years. We classified household income into "low," "medium," and "high" categories based on The Pew Research Center classification of "Middle-Income" being 67%-200% of the median household income for the region and year, which also precluded the need to adjust for inflation across

years [51]. We determined significant differences for education, race, and income class across HOLC categories and years using a generalized linear model (glm) with interactions between HOLC category and years (and income class for income demographics). Models were fit with a quasibinomial distribution to account for the potential clustering of data within HOLC categories, with independent variables of HOLC category and year, and the demographic proportion being our dependent variable. Because education, race, and income are known to be correlated with each other, we modeled each demographic factor independent from each other. This resulted in five unique models: HOLC category and Year effects on non-White residents; college graduates; and low, middle, high income households. We determined significance of interactions between demographic data and HOLC category through p-values with an alpha of 0.05.

## Analysis of gentrification

We examined connections to land use and patterns of gentrification by using a previously developed "gentrification index" that determines whether a Census tract has gentrified or not between the years of 2000 and 2019 [52,53]. A census tract is determined to be eligible for gentrification if its median income and home value was within the bottom $40^{th}$ percentile compared to tracts within the same metro area during the year 2000. This eligibility identifies groups of lower-income neighborhoods with below average household values. A neighborhood was then determined to be "gentrified" if it met all the following criteria: [1] the median home value had increased when adjusted for inflation within the study period (in this case 2000–2019), and [2] the inflation-adjusted median home value placed the tract in the top one-third of all tracts within the metro area in the year 2019, and [3] the increase in college-educated adults (holding a minimum of a bachelor's degree) was also in the top one-third of tracts within the same metro area in the year 2019. Using the Tabulate Intersection tool in ArcMap 10.8, we determined the percentage of each HOLC area covered by gentrified tracts, tracts eligible for gentrification, and tracts ineligible for gentrification. To determine demographic influence on total gentrified area for each HOLC category, we conducted linear regressions using the percentage of non-White population as an independent variable. Because the gentrification index itself is calculated using education and income data, we did not use those demographic inputs as independent variables in our analysis of gentrification drivers. We completed this analysis using the original methodology of the "Gentrification Index" from Freeman [52], the Longitudinal Tract Database developed by Logan et al. [54], and Python code developed by Vo [55]. All statistical analysis was completed in R Version 4.2.2 [56].

## Results

### Land-use diversity

We found HOLC D categories exhibited a greater diversity of land-use classes in 1937, and it continued to have higher diversity of land-use classes over time compared to HOLC categories A and B (Fig. 2A). Although land use in HOLC A categories was primarily used for single-family residences in 1937, by 1997 single-family land use in HOLC A categories had both increased by the largest percentage (25%) and was the most dominant land-use class (93%) compared to other HOLC categories (Fig. 2A,Table 1).

At the other end of the spectrum, HOLC D categories not only exhibited the lowest amount of single-family residential land use in 1937 (49%), but that value decreased to 45% by 1997. In addition, in 1937 HOLC A, C, and D categories all had agricultural land use (A: 9%, C: 12%, D: 20%) (Table 1). However, by 1957 agricultural land use was not found in use

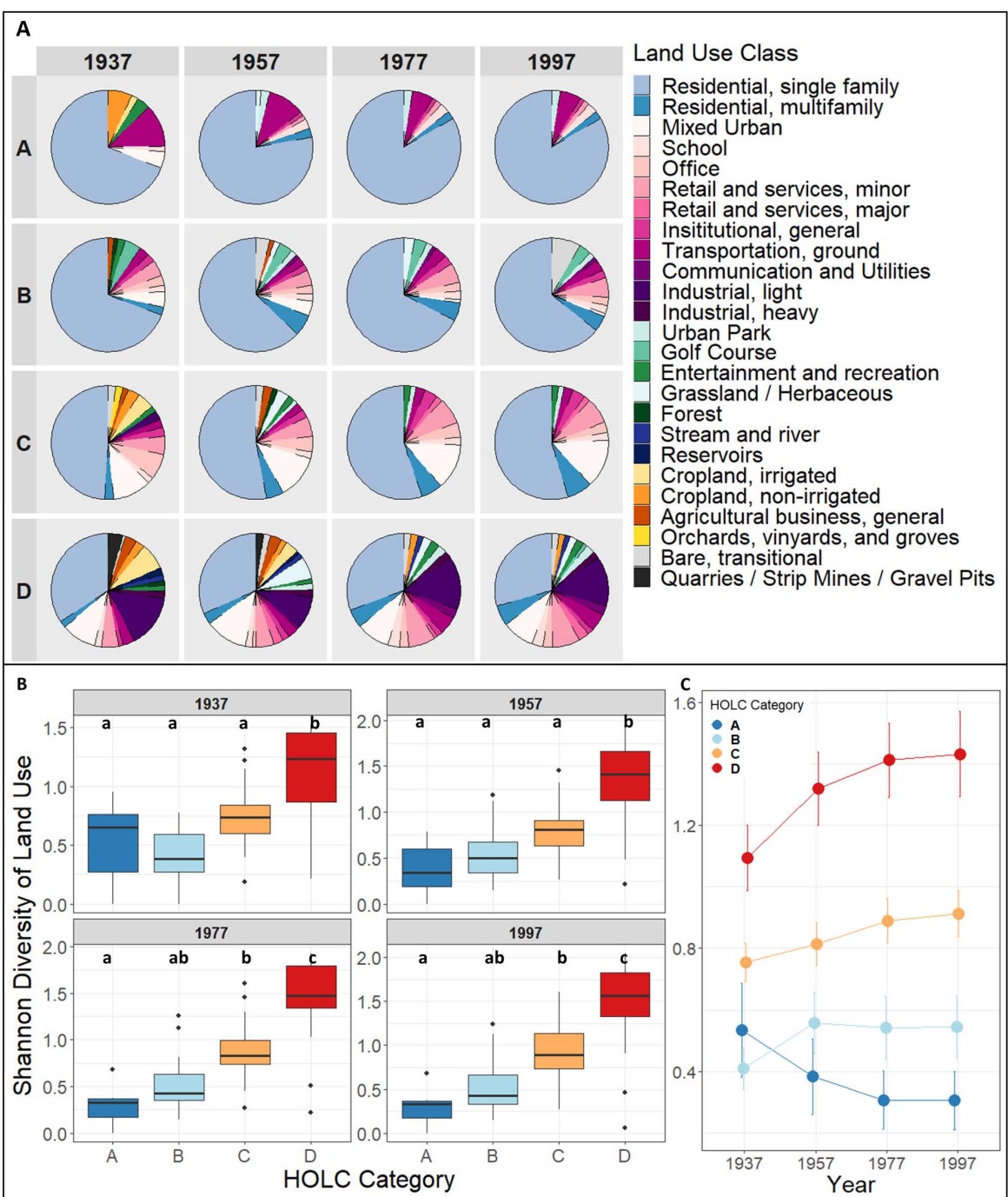

**Fig. 2. Proportions of land-use classes in Home Owners' Loan Corporation (HOLC) categories and their diversity across time.**
A. Visual description of land-use classes of HOLC categories at 20-year periods between 1937 and 1997. Percentage values are listed in Table 1. B: Boxplots comparing land-use Shannon Diversity of each HOLC category at each time-period. Significant differences between groups are represented by different letters above boxplots. The boxes represent the interquartile range (IQR) of values, the horizontal line within each box indicates the median. Outliers are shown as individual points outside the whiskers, which extend to the value nor further than 1.5 times the IQR from the box. C: Differences in Shannon Diversity visualized.

**Table 1. Land use percentages aggregated to Home Owners' Loan Corporation (HOLC) categories from 1937-1997 in Denver Colorado.**

| Land Use Type | A | | | | B | | | | C | | | | D | | | |
|---|---|---|---|---|---|---|---|---|---|---|---|---|---|---|---|---|
| | 1937 | 1957 | 1977 | 1997 | 1937 | 1957 | 1977 | 1997 | 1937 | 1957 | 1977 | 1997 | 1937 | 1957 | 1977 | 1997 |
| Agricultural business, general | 0% | 0% | 0% | 0% | 2% | 2% | 0% | 0% | 2% | 4% | 0% | 0% | 5% | 5% | 0% | 0% |
| Bare, transitional | 0% | 0% | 0% | 0% | 0% | 5% | 0% | 11% | 3% | 3% | 0% | 0% | 1% | 3% | 3% | 3% |
| Communication and Utilities | 0% | 0% | 0% | 0% | 0% | 2% | 1% | 1% | 0% | 0% | 0% | 0% | 0% | 1% | 4% | 4% |
| Cropland, irrigated | 2% | 0% | 0% | 0% | 0% | 0% | 0% | 0% | 7% | 0% | 0% | 0% | 11% | 6% | 0% | 0% |
| Cropland, non-irrigated | 7% | 0% | 0% | 0% | 0% | 0% | 0% | 0% | 5% | 0% | 0% | 0% | 4% | 3% | 3% | 3% |
| Entertainment and recreation | 4% | 0% | 0% | 0% | 2% | 0% | 0% | 0% | 2% | 3% | 3% | 3% | 2% | 2% | 4% | 3% |
| Forest | 0% | 0% | 0% | 0% | 1% | 0% | 0% | 0% | 0% | 2% | 0% | 0% | 2% | 0% | 0% | 0% |
| Golf Course | 0% | 0% | 0% | 0% | 5% | 5% | 4% | 4% | 0% | 0% | 0% | 0% | 0% | 0% | 0% | 2% |
| Grassland/ Herbaceous | 0% | 2% | 0% | 0% | 0% | 2% | 4% | 0% | 0% | 5% | 0% | 0% | 0% | 10% | 4% | 3% |
| Industrial, heavy | 0% | 0% | 0% | 0% | 0% | 0% | 0% | 0% | 0% | 0% | 0% | 0% | 3% | 3% | 3% | 3% |
| Industrial, light | 0% | 0% | 0% | 0% | 0% | 0% | 0% | 0% | 3% | 0% | 0% | 0% | 22% | 15% | 23% | 22% |
| Institutional, general | 0% | 1% | 1% | 1% | 3% | 2% | 3% | 3% | 3% | 3% | 5% | 5% | 2% | 3% | 3% | 3% |
| Mixed Urban | 5% | 0% | 0% | 0% | 5% | 6% | 1% | 1% | 15% | 17% | 17% | 18% | 14% | 20% | 14% | 14% |
| Office | 0% | 0% | 0% | 0% | 3% | 3% | 3% | 0% | 10% | 6% | 5% | 4% | 0% | 2% | 4% | 4% |
| Orchards and groves | 0% | 0% | 0% | 0% | 0% | 0% | 0% | 0% | 3% | 0% | 0% | 0% | 0% | 0% | 0% | 0% |
| Quarries/ Gravel Pits | 0% | 0% | 0% | 0% | 0% | 0% | 0% | 0% | 0% | 0% | 0% | 0% | 6% | 4% | 0% | 0% |
| Reservoirs | 0% | 0% | 0% | 0% | 0% | 0% | 0% | 0% | 0% | 0% | 0% | 0% | 3% | 2% | 0% | 0% |
| Residential, multifamily | 0% | 3% | 3% | 3% | 3% | 8% | 6% | 6% | 3% | 7% | 8% | 9% | 3% | 5% | 8% | 9% |
| Residential, single family | 68% | 89% | 93% | 93% | 78% | 85% | 85% | 85% | 67% | 74% | 71% | 70% | 49% | 51% | 46% | 45% |
| Retail and services, major | 0% | 0% | 0% | 0% | 0% | 0% | 0% | 0% | 0% | 0% | 2% | 2% | 0% | 5% | 1% | 5% |
| Retail and services, minor | 0% | 1% | 2% | 2% | 5% | 6% | 7% | 7% | 7% | 8% | 10% | 10% | 7% | 8% | 11% | 12% |
| School | 2% | 3% | 3% | 3% | 2% | 3% | 3% | 3% | 2% | 3% | 3% | 3% | 3% | 3% | 5% | 5% |
| Stream and river | 0% | 0% | 0% | 0% | 0% | 0% | 0% | 0% | 0% | 0% | 0% | 0% | 2% | 2% | 2% | 2% |
| Transportation, ground | 12% | 12% | 7% | 7% | 3% | 3% | 4% | 4% | 3% | 3% | 3% | 3% | 5% | 5% | 8% | 8% |
| Urban Park | 0% | 3% | 3% | 3% | 0% | 2% | 2% | 2% | 0% | 2% | 2% | 2% | 0% | 3% | 4% | 4% |

within HOLC A, with a corresponding increase in the area covered with single-family homes. In HOLC D, agricultural land did not drop below 10% until 1977, by which time single-family residential area had decreased, while small increases occurred in multiple commercial and industrial land uses. When comparing residential land-use data from 2018 (40%) to that from 1997 (19%), we found an increase in high-income households was positively correlated with an increase in residential land use in only HOLC D neighborhoods (S1 Fig).

Overall, land-use diversity within each HOLC category followed distinct patterns. For all decades of land-use diversity analysis we found the HOLC category had a significant effect on Shannon Diversity (1937: F = 9.44, DF = 3, p-value < 0.0001; 1957: F = 13.67, DF = 3, p-value < 0.0001; 1977: F = 18.53, DF = 3, p-value < 0.0001; 1997: F = 16.23, DF = 3, p-value < 0.0001). In 1937 only HOLC D was significantly different from all other HOLC categories, but by 1977, both HOLC categories C and D were significantly distinct from each other, and from HOLC A (Fig. 2B). HOLC D categories exhibited an increase in land-use alpha diversity every 20 years, while HOLC A displayed a decreasing pattern in land-use diversity over the same period (Fig. 2C). When analyzing zoning data, we found areas formally designated HOLC-D had significant higher percentages of areas zoned for commercial, industrial, and mixed-use compared to former HOLC A areas (S2A Fig). Furthermore, we found the HOLC C and D categories have a significantly greater total of zoning changes from 1955 to 2019, and that these changes have occurred incrementally during that entire period (S2B Fig and 2C).

## Demographics

From 1940 onward, patterns of race and education across HOLC categories were similar to patterns of land-use diversity (Fig. 3A and B). The percentage of non-Whites both increased with year and HOLC Category, with an interaction effect between those two factors (HOLC category: F = 20.84, DF = 3, p-value < 0.001; Year: F = 34.31, DF = 4, p-value < 0.001; HOLC * Year: F = 1.92, DF = 12, p-value = 0.03). The percentage of non-Whites increased from HOLC A to HOLC D categories (Table 2), increased overall in Denver from 1940 to 2019,

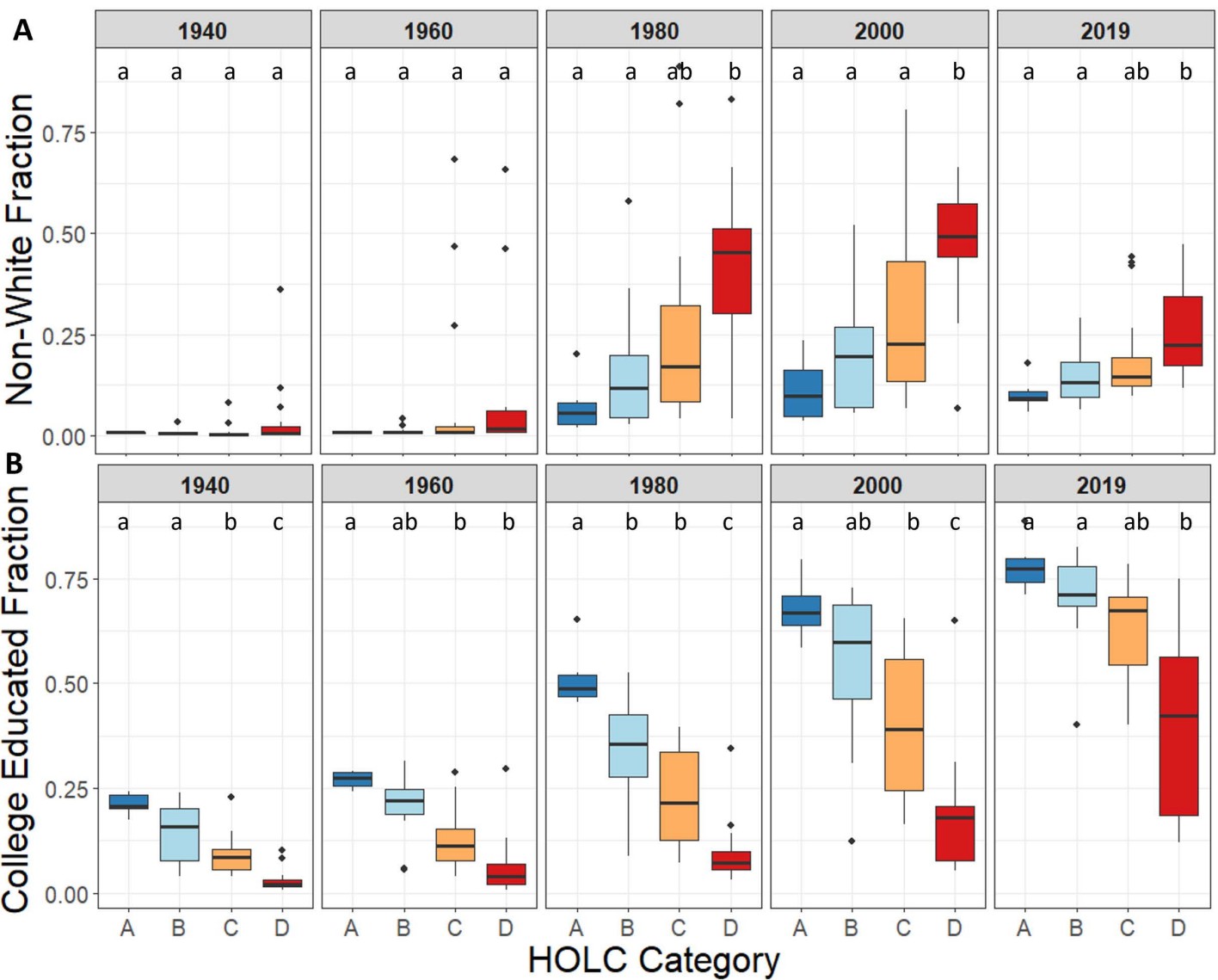

**Fig. 3. Patterns of racial and educational demographics of Denver residents within Home Owners' Loan Corporation (HOLC) categories over eight decades.**
*A: Boxplots comparing the non-White proportion of each HOLC-categorized area at each period. B: Boxplots comparing the college-educated proportions of each HOLC-categorized neighborhood at each period. The boxes represent the interquartile range (IQR) of values, the horizontal line within each box indicates the median. Outliers are shown as individual points outside the whiskers, which extend to the value nor further than 1.5 times the IQR from the box. Significant differences between HOLC categories are represented by different letters above boxplots. We determined significance through a generalized linear model with interactions between HOLC category and years. We obtained data from the U.S. decennial Census through IPUMS and the 2019 American Community Survey [10,50].*

and was significantly higher in HOLC D than HOLC A and B from 1980 to 2019. The proportion of college-educated individuals increased as a factor of year, of HOLC category, and an interaction effect between the two factors (HOLC category: F = 72.09, DF = 3, p-value < 0.001; Year: F = 155.09, DF = 4, p-value < 0.001; HOLC * Year: F = 3.17, DF = 12, p-value < 0.001). This means that while the college-educated populations decreased from HOLC A to D categories and the overall number of college-educated populations also increased from 1940 to 2019, the size of that increase was significantly smaller in HOLC D areas in 1980, 2000, but significantly larger in 2019, compared to other HOLC categories. Within decades, HOLC categories did not differ statistically in the proportions of non-White population in 1940 or 1960, although by 1980, HOLC D categories had statistically significant higher non-White proportions. HOLC D also had significantly lower college-educated populations compared to HOLC A for the entire study period.

The proportion of households in different income classes varied across year and HOLC category (Fig. 4). In low- and high-income households, we observed a significant effect of HOLC category and year on proportion of low-income households, but no significant interaction between HOLC category and year (Low Income - HOLC category: F = 59.87, DF = 3, p-value < 0.001; Year: F = 94.45, DF = 3, p-value < 0.001; HOLC * Year: F = 1.66, DF = 9, p-value = 0.10. High Income - HOLC category: F = 66.42, DF = 3, p-value < 0.001; Year: F = 48.16, DF = 3, p-value < 0.001; HOLC * Year: F = 1.28, DF = 9, p-value = 0.252). For middle-income households, we observed a significant interaction between HOLC category and year (HOLC category: F = 3.55, DF = 3, p-value = 0.016; Year: F = 45.12, DF = 3, p-value < 0.001; HOLC * Year: F = 5.64, DF = 9, p-value < 0.0001). During 1980, the HOLC C and D category increased in the proportion low-income households significantly more than HOLC A or B (HOLC C – Year 1980: p-value = 0.005; HOLC D – Year 1980: p-value = 0.001). Within decades, in 1960, high- and middle-income households displayed differences between HOLC categories, with HOLC C and D categories having significantly lower proportions of middle and high-income households, and middle-income households. Although there was no difference between low-income household proportion in 1960, in 1980 and the decades following, we found significantly higher proportions of low-income households in HOLC D categories (Fig. 4).

The directions and magnitude of proportional changes in our demographic metrics varied depending on year and variable. While every 20-year period displayed an increase in the college-educated proportion of population, the magnitude of change was generally always largest in HOLC D, decreasing in magnitude to HOLC A, with the largest magnitude of change in the latest period. The non-White proportion of population also increased across all HOLC categories up to the year 2000, with very large increases from 1940 to 1980, where there was a proportional decline in non-White residents in HOLC B, C, and D categories, with HOLC D experiencing the largest proportional decline. Income metric patterns also varied by variable. The proportional

**Table 2. Percent changes to demographic metrics in Home Owners' Loan Corporation (HOLC) categories. Education and racial data start in 1940 and income data start in 1960.**

| Demographic Metrics | A | | | | B | | | | C | | | | D | | | |
|---|---|---|---|---|---|---|---|---|---|---|---|---|---|---|---|---|
| | 1940–1960 | 1960–1980 | 1980–2000 | 2000–2019 | 1940–1960 | 1960–1980 | 1980–2000 | 2000–2019 | 1940–1960 | 1960–1980 | 1980–2000 | 2000–2019 | 1940–1960 | 1960–1980 | 1980–2000 | 2000–2019 |
| College Educated Fraction | 29% | 90% | 33% | 15% | 47% | 94% | 69% | 48% | 35% | 111% | 84% | 78% | 56% | 245% | 102% | 147% |
| Non-White Fraction | 2% | 1463% | 71% | 22% | 201% | 2408% | 48% | -5% | 1279% | 2016% | 42% | -21% | 1091% | 3626% | 22% | -41% |
| High-Income Households | na | -43% | 86% | 5% | na | -62% | 220% | 49% | na | -69% | 296% | 87% | na | -76% | 612% | 58% |
| Medium-Income Households | na | 14% | -26% | 2% | na | -5% | -15% | -7% | na | -36% | 17% | 0% | na | -34% | 28% | 7% |
| Low-Income Households | na | 109% | -36% | -5% | na | 172% | -36% | -20% | na | 147% | -30% | -23% | na | 156% | 3% | -18% |

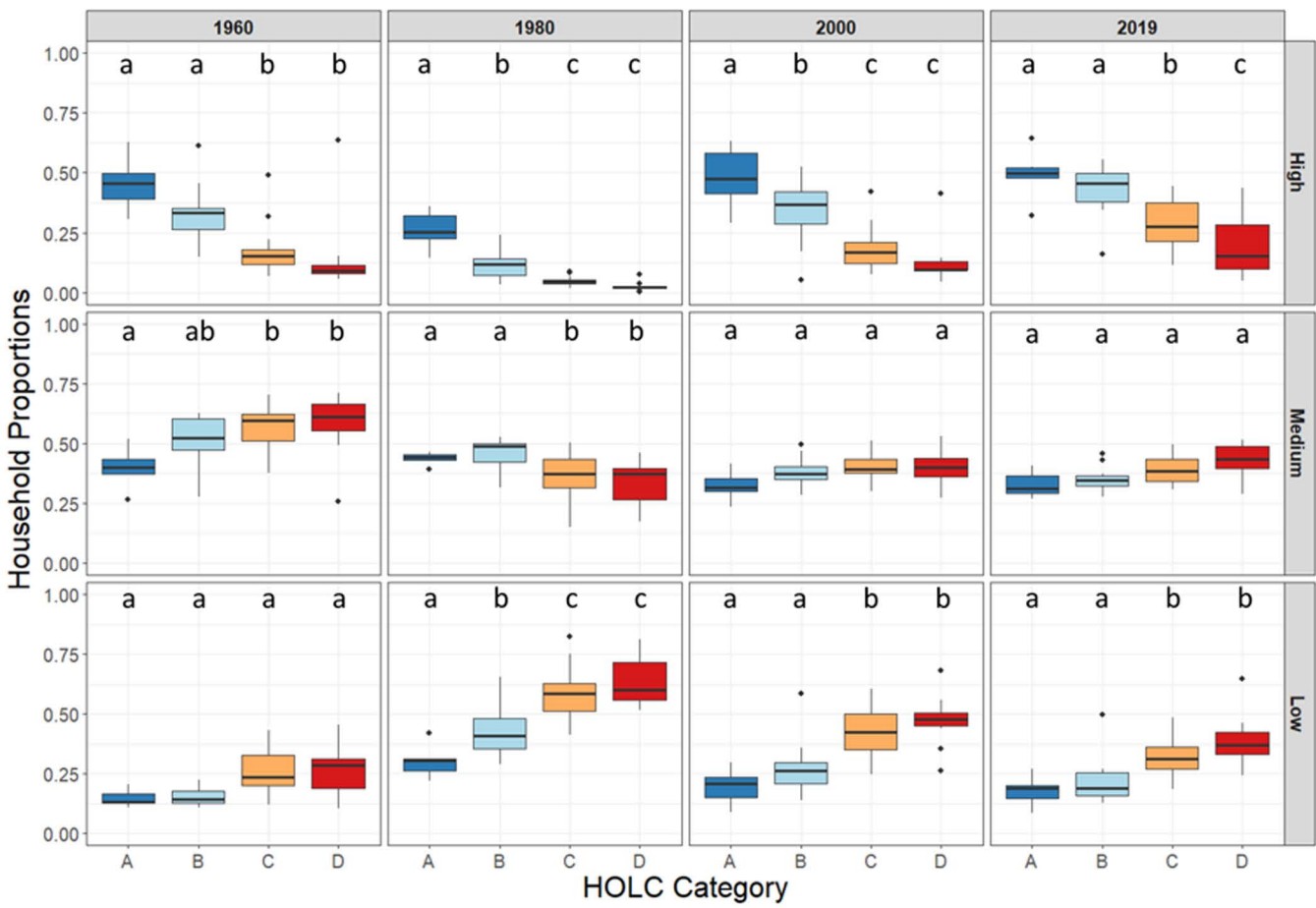

**Fig. 4. Patterns of income of Denver residents within Home Owners' Loan Corporation (HOLC) categories over eight decades.** *A:* Boxplots comparing the proportion of low-, medium-, and high-income households each HOLC-classified neighborhood at each period. The boxes represent the interquartile range (IQR) of values, the horizontal line within each box indicates the median. Outliers are shown as individual points outside the whiskers, which extend to the value nor further than 1.5 times the IQR from the box. Significant differences between groups are represented by different letters above boxplots. We obtained data from the U.S. decennial Census through IPUMS and the 2019 American Community Survey [10,50].

change to high-income households maintained the same direction of change across all HOLC categories for each period, with increases in high-income households occurring steadily after 1980 (Fig. 4 and Table 2). For the medium- and low-income variables, the patterns varied but magnitude differences across HOLC categories remained relatively low. These patterns indicate that from 1940 onward, Denver has attracted college-educated and racially diverse residents across the city, with patterns in income following similar macro-citywide patterns. However, in the last 20 years, the HOLC C and D categories have seen the largest decrease in diverse populations, as well as the largest increases in college-educated residents and high-income households.

## Gentrification

According to our "gentrification index," between 2000 and 2019, 100% of HOLC A areas were ineligible for gentrification, with decreasing areas of ineligibility occurring in HOLC's B through D. HOLC D contained significantly more area eligible for gentrification in 2019 compared to all other HOLC categories, although no difference was found between HOLC categories and percent area gentrified (Fig. 5A). When examining racial characteristics as a

driver of gentrification within neighborhoods eligible for gentrification, higher proportions of non-White population predicted higher percentages of gentrification only in HOLC C categories (p-value = 0.001, Fig. 5B).

## Discussion

Our aim was to address the gaps in knowledge regarding the legacy of social and land-use patterns varied across HOLC categories over the past century. By pairing unique high-resolution land-use data with demographics from the U.S. Census, we show how historical redlining not only illustrated differing patterns of land use and demography prior to the creation of Denver's HOLC map, but also how these patterns continued to evolve over time. Certain differences between HOLC A and HOLC D areas in Denver became statistically significant both in the years prior to redlining and in the decades following, as we observed significant differences of land-use diversity in 1937, but not until 1980 (after the practice of redlining was made illegal by the Fair Housing Act of 1968) for demographics of race and low-income status. Also, the evidence of significant segregation in land-use composition, education, and high-income residents dating to the creation of Denver's HOLC map adds weight to the argument that segregation-based lending practices incorporated pre-existing segregation and reinforced it

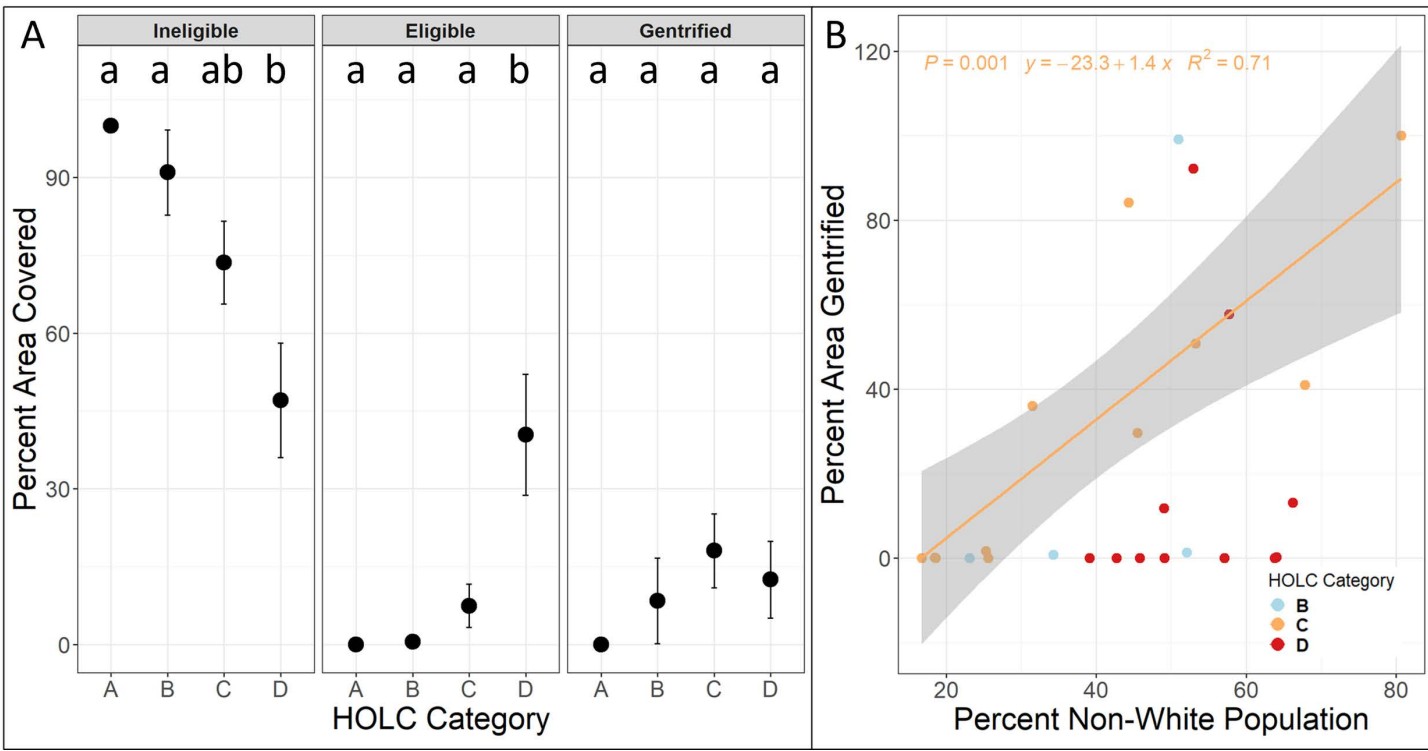

**Fig. 5. Gentrification metrics for Home Owners' Loan Corporation (HOLC) category and relationship to race.** *A: The 2019 area percentage in HOLC category that is either ineligible for gentrification (household income and median home value were above the 40th percentile within the metro area in 2000), eligible for gentrification (household income and median home value were in the bottom 40th percentile within the metro area in 2000), or gentrified (eligible tracts from 2000 experienced an increase in home value by 2019, and median income and percent college-educated residents were within the top one-third for the metro area). The middle point corresponds with the data mean and whiskers corresponds to the standard error for data within the HOLC category and gentrification variable. A significant difference between groups is represented by different letters. B. The relationship between the percentage of non-White residents in the year 2000 and the percent area gentrified in all HOLC areas eligible for gentrification in 2019. Shaded area represents the estimated standard error. We derived data from the U.S. decennial Census through IPUMS, the 2019 American Community Survey and the Longitudinal Tract Database [10,50,54].*

after these practices were made illegal [7]. These results illustrate the multi-decadal legacy of urban segregation and the potential reinforcement of that segregation through discriminatory housing policies and neighborhood reinvestment. However, between 2000 and 2019, shifts in some of the demographic patterns emerged in demographics and land use, indicating how long-term patterns can be rapidly changed by gentrification.

## Drivers of land-use diversity

Various regulatory mechanisms—such as zoning, restrictive covenants, barriers to funding, and special districts—influence land-use changes [57–59]. Unraveling the cause and effect of these mechanism is beyond the scope of this paper; however, the HOLC boundaries provide a fixed spatial extent in which descriptive information about racial and ethnic diversity and economic indicators were standardized nationally and shed light on the pre-existing conditions of a city prior to mortgage lending practices having an impact [7,13]. Our data corroborates this: prior to 1940, an inequality of land-use diversity, higher-income households, and college-educated populations already existed across Denver's HOLC areas. Our evidence supports Hillier's 2003 hypothesis that HOLC practices are not causes of present-day inequality, but reinforced segregation through "racial steering" [13]. Further comparing HOLC areas over time, we found significant differences in the Shannon-Wiener index as early as 1937 followed by differing patterns of land-use diversity across HOLC groups in subsequent decades. HOLC A areas declined in alpha diversity after 1937, and then stabilized after 1977. However, in HOLC C and D, the land-use alpha diversity continued to increase post-1937. This change in land-use diversity is reflected in the significant amount of multiple zoning categories in formally HOLC D areas compared to formally HOLC A (S2A Fig), and the regular changes in zoning occurring in HOLC C and D between 1960 and 2020 (S2B Fig and 2C). Detailed information about directional changes in zoning ordinances (specifically knowing what the zoning was before a change), restrictive covenants, or special districts that may have affected these patterns were excluded from our analysis because they could not provide the same level of standardized descriptive information or fixed spatial extents.

Given that the drivers of land-use diversity are multi-faceted and complex, we focus on understanding the balance between the fixed or flexible nature of land use in HOLC areas over time. Although we do not have records of where FHA and HOLC loans were distributed in Denver in the 1930s, the fixed versus variable presence of single-family homes across HOLC categories indicates the lasting impact of denying low-cost mortgage insurance in HOLC D categories. Single-family residential zoning is the most fixed of any land-use type, once a single-family residential area is established, it almost always remains as single-family zoning [60,61]. For areas with detached homes, an increase in homeownership rates can create the demand for the area to be converted to single-family zoning [62,63]. Because potential homeowners HOLC D were prevented from achieving homeownership, these neighborhoods did not have the chance to establish single-family zoning and would have a higher chance of land use diversifying over time. Our results then provide evidence for the permanence of single-family residential areas: by 1957 HOLC classes A-C had increases in single family residential compared to HOLC D. Moreover, post 1957, residential land uses in HOLC A and B remained stable, whereas in HOLC D after 1957 there is a small decline of the percentage of single-family homes and a steady increase in multi-family homes and other non-residential land-use classes (Table 1). The fluctuation in single-family homes and greater overall land-use diversity within redlined districts (Fig. 2) implies HOLC D categories were more vulnerable to changes in land-use regulation. Thus, the land-use diversity in HOLC D categories appears to be the result of a combination of micro- and macro-scale influences.

The city-wide urban renewal polices during the 1960s through 1980s resulted in substantial land-use and demographic changes in downtown districts. Although the downtown districts themselves—which had relatively minimal residential land use when the Denver HOLC maps were made—were not assigned categories, the downtown is surrounded by formally HOLC D categories. DURA was created in 1958 with the aim of attracting more business to Denver's downtown and creating more public amenities within generally low-income, Hispanic neighborhoods [39]. Two of DURA's largest projects were the Skyline and Auraria projects, which involved the removal of "blighted" housing and relocation of mostly lower-income Hispanic residents, and both projects faced strong public resistance before their eventual adoption [64,65]. Although these urban renewal efforts, along with subsequent smaller downtown land-use developments during the 1980s through early 2000s (e.g., Coors Field, The Denver Performing Arts Center, Confluence Park, and The Convention Center), did not directly occupy redlined areas, the rapid redevelopment and increase of amenities within downtown attracted young educated individuals to the affordable locations in adjacent HOLC D districts [65]. Consequently, all three HOLC D areas that have gentrified surround these downtown urban renewal developments (HOLC D9, D10, and D12).

## Shifts in land-use patterns and demographics

The historical legacy of diversity of land-use composition within initially described "declining" areas of Denver may result in the development of new residential land uses to attract higher-income earners, at the possible expense of often lower-income long-term residents. Our data alone, however, does not state whether displacement is occurring or has occurred. Rather, we show how the built environment has been fixed in some areas (HOLC A/B) and flexible in others (HOLC C/D). That flexibility in built environment is seen in the loss of single-family homes in formally HOLC D neighborhoods between 1957 to 1997 and the increases in multifamily homes, light industrial, and minor retail services during the same period. Those changes in land use are mirrored by dynamic shifts in demographics, such as the proportional loss in high- and middle-income households in formerly HOLC D areas between 1960 and 1980. Then in the following two decades, formerly HOLC D areas had the largest percentage increase in high-income households and college-educated residents (Table 2). More research would be useful understand these shifts in income and demographics. Given the staying power of single-family residential zoning, it is possible that HOLC A/B categories simply cannot densify at the same rate as HOLC C/D categories due to the large proportion of single-family residential zoning. However, given the vulnerabilities of low-income and non-White communities to displacement, understanding whether these changes reflect an increase in residents overall or an exodus of low-income and non-White households is an important area for future research.

Understanding the "chicken or egg" scenario of whether urban redevelopment opportunities drive demographic shifts, or if land-use changes are taken advantage of after demographic changes occur, is an active area of study [25,26]. Moreover, studies have opined that segregation-based housing policies may have directly influenced the rise of gentrification in the last 20 years [39,66]. In Denver, we found that the land-use patterns within HOLC A categories were stable after 1960, but HOLC C and D areas had greater diversity of land use during all periods, exhibiting a larger variation of residential and urban developed classes. A similar outcome was found in post-apartheid South Africa, where the Brixton neighborhood of Johannesburg has become more ethnically diverse in the years following the end of apartheid (1996–2011), and has shown evidence of becoming a desirable place for professional investment, but at the time of analysis (2017) did not show evidence of gentrification [20].

## Land-use and demographic drivers of gentrification

The gentrification that results from established patterns of land use and demographics may take many years to become noticeable. For example, we observed a general increase in proportion of White and college-educated populations in HOLC C and D areas in 2019 (Fig. 3), and this increase in these demographics to areas HOLC initially described as "declining" or "hazardous" acts as a potential indicator of gentrification. Although we found a statistical relationship between race and gentrification in formerly HOLC C but not HOLC D areas, the fact that the formerly HOLC D category has significantly more "gentrification-eligible" area could indicate if these demographic patterns continue, more formerly HOLC D areas may become gentrified. We expect this is likely to occur because the recent finding that neighborhoods with higher incomes (like the formerly HOLC C compared to HOLC D areas) are more likely to gentrify [28]. As indicated in our results, the proportion of high-income homes has grown in formerly HOLC D areas (Fig. 4). This result, combined with the high number of gentrification-eligible areas, indicates that continued gentrification of some formerly HOLC D areas is likely.

Our gentrification analyses were somewhat limited because HOLC areas that may have gentrified before 2000 would show up as ineligible for gentrification. For example, HOLC D2 (the Cherry Creek neighborhood) was described in the HOLC documentation as "Developers and real estate men have looked forward to buying cheap houses in this area for newer development. [1]" New investment did come to Cherry Creek after the construction of the Cherry Creek Dam and the subsequent Cherry Creek Shopping Center in the early 1950s, followed by the center's $250 million revitalization in 1990 [67]. Although gentrification can occur due to redevelopment of land use and increased construction of residential housing for wealthier newcomers to a neighborhood, more affluent White individuals may also move into pre-existing, lower-cost homes in HOLC C and D categories, replacing previous residents who are lower-income and People of Color [68]. This top-down land-use development and bottom-up demographic displacement are likely occurring in tandem and reinforcing each other. We find suggestive evidence of both phenomena, for example a bottom-up demographic replacement being where a larger proportion of non-White residents indicated a greater percentage of formerly HOLC C areas to be gentrified (Fig. 5), and a potential top-down linkage between land use and demographics being the increase in higher-income households in formerly HOLC D areas correlating with an increase in residential land use (S1 Fig).

In the United States, although HOLC maps described pre-existing land-use and demographic segregation, we found that further patterns of segregation continued to develop over time but can also quickly change as observed between 2000 and 2019 during periods of rapid growth and gentrification in Denver. The patterns we observed are likely not unique to Denver. Redlining policies were put in place in many cities across the United States. Patterns of gentrification over the past 20 years have also been observed in multiple cities with a history of redlining [69]. However, with the multiple studies examining the present-day effects of redlining practices, current sociological, economic, public health, and ecological patterns resulting from redlining are not the same in all cities [70]. For example, recent work has found that in the city of Portland, Oregon, urban tree canopy cover actually decreased in former HOLC A while increasing in HOLC B – D; while over the same period in Philadelphia, Pennsylvania, tree canopy decreased over all HOLC categories [22]. Moving beyond the United States, while other nations exhibit historical and present institutionalized exclusionary housing policies [20,34], un-official discriminatory housing actions are found in cities worldwide, and can play a part in establishing and re-enforcing land-use and demographics patterns of discrimination [35,71].

## Limitations

The purpose of our study and research question was to resolve uncertainty around HOLC's influence on land use and demographic changes over the past 80 years and examine if those patterns help explain where gentrification occurred in the city of Denver. This is a novel contribution to research on redlining because it used a unique time-series dataset of high-resolution land use for the city of Denver. This highlights an important limitation: land-use findings cannot be replicated for other cities without the presence of such a dataset. Data availability is often a limitation for redlining studies aiming to track change through time, and such studies often require a substantial amount of work just in recovering historical data. Or, in the case of including multiple drivers to land-use change, finding and digitizing additional land-use regulation data such as zoning, racial covenants, or special districts that may be available for only certain time periods is challenging. On the other hand, our use of demographics highlights a publicly available source of time-series data (IPUMS National Historic Geographic Information System at the University of Minnesota (www.nhgis.org)) that this study has shown is significantly tied to pre-HOLC segregation and to the legacies that followed over the decades [10]. Thus, this research presents a possible path forward for similar studies in other cities.

## Conclusion

We came to the study with the goal of finding patterns in the multidecadal data of land use and demographics that could help explain present day patterns of gentrification in a U.S. city. By using the HOLC boundaries, we were able to track how the segregation in land-use diversity, housing, and demographics was established prior to redlining practices in Denver, but continued to grow in the following 80 years. Redlining was a key aspect of this divergence in patterns, but urban renewal policies and projects of the mid to late 21st century also reinforced land-use and demographic shifts. These projects, along with urban redevelopment brought more educated and wealthy individuals to formally HOLC C and D areas, which may have been able to accept these demographic changes due to diverse and changing land use patterns. Although our study does not address the issues of demographic replacement caused by gentrification, the patterns we observe do answer questions about how present-day patterns in urban land cover and land use can both develop over many decades, but also be primed to have large shifts towards gentrification if certain conditions are met. Moreover, although the discriminatory housing policies of the 21st century are not unique to Denver, Colorado, or even the United States, our study was able to highlight uniquely city-specific drivers of change from the past decades. Future studies examining redlining's impacts could consider how discriminatory patterns develop over many decades as well as examining what the present-day affects may be.

## Disclaimer

Any use of trade, firm, or product names is for descriptive purposes only and does not imply endorsement by the U.S. Government.

## Supporting information

**S1 Fig. Comparison of increases in residential land use within Home Owners' Loan Corporation (HOLC) categories between 1997 and 2018, and increases in high-income households within the same neighborhoods over the same period in Denver, Colorado.** The 1997 and 2018 land-use data are from two different sources (1997: Drummond et al.

2019; 2018: Chen et al 2022) with differing classification methodologies. Only significant relationships have statistics and regression lines displayed. Shaded area represents the estimated standard error.
(TIF)

**S2 Fig. The influence of zoning ordinances, and their changes on land use configuration across former Home Owners' Loan Corporation (HOLC) categories in Denver, Colorado.** A: A comparison of the Zoning Class Percentages for six difference zoning designations across HOLC categories in the year 2020. B: The total area of parcels in the year 2020, which experienced a zoning change from 1955 to 2020. C: The total sum of area which experienced a zoning change across HOLC categories, aggregated to the decadal scale. The middle point corresponds with the data mean and whiskers corresponds to the standard error for data within the HOLC category and zoning variable. Significant differences between groups are represented by different letters above boxplots.
(TIF)

## Author contributions

**Conceptualization:** Peter C. Ibsen, Anna Bierbrauer, Kenneth J. Bagstad, Jay E. Diffendorfer.

**Data curation:** Anna Bierbrauer, Lucila M. Corro, Zachary H. Ancona, Mark Drummond.

**Formal analysis:** Peter C. Ibsen, Anna Bierbrauer, Lucila M. Corro, Zachary H. Ancona.

**Investigation:** Peter C. Ibsen, Anna Bierbrauer.

**Methodology:** Peter C. Ibsen, Anna Bierbrauer, Zachary H. Ancona, Mark Drummond, Jay E. Diffendorfer.

**Project administration:** Peter C. Ibsen, Jay E. Diffendorfer.

**Resources:** Anna Bierbrauer, Kenneth J. Bagstad.

**Supervision:** Kenneth J. Bagstad, Jay E. Diffendorfer.

**Validation:** Peter C. Ibsen.

**Visualization:** Peter C. Ibsen.

**Writing – original draft:** Peter C. Ibsen, Anna Bierbrauer.

**Writing – review & editing:** Peter C. Ibsen, Anna Bierbrauer, Lucila M. Corro, Zachary H. Ancona, Mark Drummond, Kenneth J. Bagstad, Jay E. Diffendorfer.

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
