## [Decision Letter · Decision Letter 0]

3 Sep 2024

PONE-D-24-05471Land use and socioeconomic time-series reveal legacy of redlining on present-day gentrification within a growing United States city.PLOS ONE

Dear Dr. Ibsen,

Thank you for submitting your manuscript to PLOS ONE. After careful consideration, we feel that it has merit but does not fully meet PLOS ONE’s publication criteria as it currently stands. Therefore, we invite you to submit a revised version of the manuscript that addresses the points raised during the review process. These comments are located at the bottom of this message.

We look forward to receiving your revised manuscript.

Kind regards,

Christopher A. Lepczyk

Academic Editor

PLOS ONE

Journal Requirements:

"NO"

4. We note that your Data Availability Statement is currently as follows: [All relevant data are within the manuscript and its Supporting Information files]

5. We note that [Figure 1] in your submission contain [satellite] images which may be copyrighted. All PLOS content is published under the Creative Commons Attribution License (CC BY 4.0), which means that the manuscript, images, and Supporting Information files will be freely available online, and any third party is permitted to access, download, copy, distribute, and use these materials in any way, even commercially, with proper attribution. For these reasons, we cannot publish previously copyrighted maps or satellite images created using proprietary data, such as Google software (Google Maps, Street View, and Earth). For more information, see our copyright guidelines: http://journals.plos.org/plosone/s/licenses-and-copyright.

6. Please include a new copy of Table xxxx in your manuscript; the current table is difficult to read. Please follow the link for more information: https://blogs.plos.org/plos/2019/06/looking-good-tips-for-creating-your-plos-figures-graphics/

Additional Editor Comments:

Associate Editor:

General Comments

My apologies for the long delay in returning a decision on the manuscript. We went through an excessively large number of reviewers to obtain two qualified individuals to assess the work. As you will note, both reviewers found merit in the work and appreciated the level of analysis done. The main item from the reviews that warrants further consideration is the use/inclusion of zoning data. If zoning data are available for part or all of the time series, it would greatly aid your analyses and should be pursued. If these data are unavailable, then simply note that in a rebuttal. Finally, I would suggest including a paragraph in the Discussion that clearly describes all the limitations of the work.

Specific Comments

Abstract. Include main research questions/hypotheses.

L42. Change greenspace to green space here and throughout the ms.

L46. One point that has come up that may want to note is that redlining was not the singular factor that has been recently described, but one of several that may co-occur, etc. You mostly note this point already, but may touch upon a bit more that it isn’t just a simple classification that explains the findings of other studies. Likewise, this point also relates to the reviewer’s comment about zoning.

L101-129. Much of this text can be moved to Methods or deleted. L101 should start the final paragraph that lays out the overarching goal/question followed by the specific questions/hypotheses/objectives.

L130. I would suggest revising this first sentence a bit to indicate the gap in knowledge that you are seeking to answer.

L174. Please note here or elsewhere in the Methods what datum and projection were used in your geographic analyses.

L192. How are you dealing with different resolutions of the datasets?

L215. Revise to ‘This approach…’

L223. Did you evaluate collinearity amongst terms first? If not, please do. Also, did you have to transform any of the data? Finally, are you just using p-values here or AICc?

L241. Here or elsewhere in Methods indicate the statistical software you used for analyses.

L244. I would suggest cutting this sentence. Results here should be very succinct. You found x relationship, with y parameters being important.

Fig. 2 legend. Delete ‘We determined significance through a generalized linear model with interactions between HOLC category and years.’ Also, no description of panel B is noted.

L276. Can revise to just indicate result, do not need to describe the statistical approach used as that was presented in the Methods.

L277. Provide the full statistical results (F = xxx, DF = x,y, p = z.zz) here and elsewhere in Results. Also, delete the sentence that starts at this line as you already explained post hoc approach in Methods.

L290. See previous comment on including detailed stats.

L301, L315, L358. Provide the statistical details of the analyses here.

Fig. 3 legend. Delete ‘We determined significance through a generalized linear model with interactions between HOLC category and years.’

Fig. 4 legend. Delete ‘We determined significance through a generalized linear model with interactions between HOLC category, income class, and years.’

L354. Move this item to Methods.

Fig. 5 legend. Delete ‘We determined significance through a generalized linear model with interactions between HOLC category and years.’ And ‘Regression line equations and p values by HOLC category are provided.’

L374. Can cut most of first two sentences. Begin with answers to your questions.

Reviewers' comments:

Reviewer's Responses to Questions

**Comments to the Author**

1. Is the manuscript technically sound, and do the data support the conclusions?

Reviewer #1: Partly

Reviewer #2: Yes

2. Has the statistical analysis been performed appropriately and rigorously? 

Reviewer #1: Yes

Reviewer #2: Yes

3. Have the authors made all data underlying the findings in their manuscript fully available?

Reviewer #1: Yes

Reviewer #2: Yes

4. Is the manuscript presented in an intelligible fashion and written in standard English?

Reviewer #1: Yes

Reviewer #2: Yes

5. Review Comments to the Author

Reviewer #1: The principal claim of this article is succinctly summarized in the title: “Land use and socioeconomic time-series reveal legacy of redlining on present-day gentrification within a growing United States city,” but the study asks and answers much more far-reaching claims than this. And, the study effectively qualifies the specific role of “redlining,” meaning the impact of the 1938 HOLC Security Maps as a causal factor in this overall process. Instead, the authors use the HOLC categories as a snapshot of the land use and demographic conditions at the starting-point in their longitudinal study. The goal at the heart of the study is to track long-term changes in land use diversity and demographic areal composition from 1937 to 2019. While the long-term correlations between HOLC categories (developed originally and used by lending agencies thereafter for the purpose of qualifying applicants for mortgages) and downstream social inequality, urban amenities, environmental injustice, and other factors has been well-documented now in numerous studies amply cited in this study, the key innovation here is the use of an allegedly unique—for Denver-- high-resolution land-use data series (Drummond et al 2019) , for Denver in the years 1937, 1957, 1977, and 1997. This data series allows the authors to a) set a benchmark of land use prior to any possible impact of the HOLC maps, and b) track changes in 20-year intervals between the onset of the redlining era and pour contemporary, particularly the first two decades of the 21st century.

In the authors’ own words, “The spatial patterns of this connection between the built environment, past discriminatory policies, and the potential vulnerability to neighborhood gentrification have been connected from the present to the HOLC boundaries (30), but to understand how these spatial patterns emerged over time, time series data is needed. (96-101). The authors address three specific questions: “1. How have patterns in land use and demographics developed and diverged in formally categorized HOLC areas across Denver from 1937 to 2019? 2. How has the diversity of demography in Denver neighborhoods connected to HOLC areas changed over the same period? 3. How have patterns of inequality in land use and demographics from the previous decades influenced rapid present-day changes in Denver’s demographics and resulting neighborhood-specific patterns in gentrification?” (129-138).

Overall the authors are successful in answering all three of these questions, but there is a major factor that they both recognized and fail to include in their analyses, which calls into question the whole rationale for depending on HOLC classifications in the first place: Denver’s zoning ordinances.

The authors do show statistically significant patterns in which the land-use diversity was reinforced in HOLC category D areas across the entire time period, which residential, single-family land use also increased in areas A and B. Proportions of college-educated, white, and higher-income households were all significantly lower for HOLD D areas than for HOLC-A areas over time. Given the goals of the study, a significant achievement is that the authors are able to show significant changes within the overall trends from 1937-2019 in two time-periods: 1) a significant impact of urban renewal policies and projects during the 1960-1980 interval, and 2) the impact of gentrification in areas previously disadvantaged by the longer-term trends, in the early 20th century period.

While all of this is very solid statistically, this reviewer believes that the authors made a critical methodological mistake by not identifying and including the geographic boundaries of the city’s zoning ordinances, along with the land-use restrictions codified by those ordinances for those geographic boundaries, as independent variables in addition of the HOLC boundaries. The HOLC boundaries, after all, were established for the single purpose of evaluating the loan-security risk of residential properties. Thereafter, they were used or adapted by banks for decades, but only for the purpose of home loan qualifications. The principal innovation of this study is to use the Drummond, et al land-use maps, which capture all categories of land use. The authors use these maps primarily along the dimension of “diversity,” which is essentially bi-variate in their usage: they seek to distinguish high-diversity areas (where industrial and other non-residential uses are concentrated), and low-diversity, where primarily residential areas are concentrated. Zoning ordinances were invented in the early 20th century to exclude industrial and other commercial uses from certain residential areas. (and also to optimize industrial development). They did not exclude residential development within industrial/commercial areas, but they did exclude industrial/commercial development in areas newly developed to be exclusively residential. This all happened concurrent with the rise of racial restrictions, usually to exclude non-whites from those same all-residentially zoned areas. While this reviewer is not familiar with Denver’s specific zoning ordinances, based on their use in cities with which the reviewer does have familiarity, it is nearly certain that they were used to exclude land use “diversity” from residential areas over this long period. It is also likely that they were revised significantly over the years. It is also certain that zoning ordinances have a direct impact on land use, while HOLC categories have only an indirect impact. And, it is very well known that residential property values are directly impacted by proximity to nuisances typical of industrial and commercial land uses. It is likely, therefore, that the exclusivity of HOLC areas A and B, and the land use diversity of areas C and D, can be explained entirely by the regulatory effect of these ordinances. The word “zoning” only appears three times in this manuscript (95, 403, 416), but in each of these passages the authors acknowledge the importance of zoning ordinances.

The authors laudably avoid assigning HOLC maps a singular causal role in the longer temporal patterns of land use and demographic spatial inequality. They show empirically that the inequalities of land use and demographic geography pre-existed the HOLC maps, and that changes occurred long after their implementation. This reviewer believes that the failure to consider the role of zoning geographies across this period is a major oversight. While the HOLC maps have come to stand for a long-term inscription of inequality in urban landscapes, the authors needn’t have tied their study so closely to those maps which, as they acknowledge, do not even cover downtown areas. The status quo of land use and social demography in 1937 and subsequent years could have been classified by an independent set of area units, and tracked over time for long-term patterns. The use of zoning geographies would have gone a long way toward helping to explain those patterns, as city planning authorities were required to certify all building project according to these maps—and not to the HOLC maps. This reviewer does not recommend removing the HOLC geographies from the study—far from it: the inclusion of the HOLC geographies and categories should still be an important set of variables in the study. But they should be used in conjunction with the explanatory power of the zoning geographies. This approach will a) give a much more accurate view of the long-term urban developmental historical process (recognizing the factor most likely to shape land use), and b) help to advance the latest state of the art in understanding the impact of the HOLC maps. My recommendation, therefore, does not imply scrapping the entire approach of this study. Instead, it suggests the addition of mapping layers with associated attributes, most probably chosen to coincide with the four land-use data points of 1937, 57, 77, and 97, and re-running their analyses to include the independent variable of land use restrictions on every space in the study.

Reviewer #2: This research analyzed changes in land use and socioeconomic factors, considering the impacts of historical redlining, to identify current patterns of gentrification in Denver, USA. They used boxplots and statistical analysis to test three research questions developed in this research. Given the increasing academic interest in the lingering impacts of redlining on urban dynamics, the findings here are noteworthy and provide some implications. Before publication, the manuscript needs to address the following issues.

Major issue

Introduction

1. Although the authors noted some critiques of using HOLC data in the first paragraph (e.g., over-causality), it would be beneficial to also address another criticism, such as the skepticism regarding the actual impact of HOLC (check this:

https://doi.org/10.1177/0096144203029004002).

2. More recently published papers should be cited in lines 88-90.

https://doi.org/10.1016/j.landurbplan.2024.105019

https://doi.org/10.1016/j.landurbplan.2024.105028

https://doi.org/10.1080/13549839.2024.2380854

3. The first and second research questions should be revised for clearer distinction. Currently, they have some overlap.

Methods

1. The authors used the Shannon-Wiener index (H') to calculate the compositional features of land use. Are there other metrics or methods that can assess the configurational aspect of land use? Even if two districts have similar compositional attributes, their configurational characteristics may differ significantly, and these can provide some information.

2. Why do the authors use a quasi-binomial distribution to fit their models? Providing a rationale for this choice would strengthen the paper.

3. I wonder whether the gentrification index was developed by the authors or adapted from other studies. If the authors developed it, why was "the bottom 40th percentile" chosen as a criterion? In addition, the phrase "1 of the gentrification metric (in line 237)" is confusing: so tracts eligible for gentrification satisfy at least one of the four criteria or only the first criterion?

Results

1. The findings described in lines 349-352 were difficult to verify through Fig. 5A. I recommend revising both the figure and the manuscript.

2. How many observations were used to estimate each statistical model in Fig. 5b (supplementary figure also)? Very few samples appear in each HOLC category, making it difficult to calculate statistical significance.

Discussion

1. More references should be used to compare or support this research's findings. Papers addressing regions beyond Denver could offer valuable implications.

2. Although some limitations are mentioned at the end of the Discussion, additional methodological and conceptual limitations should be provided.

6. PLOS authors have the option to publish the peer review history of their article (what does this mean? ). If published, this will include your full peer review and any attached files.

**Do you want your identity to be public for this peer review?** For information about this choice, including consent withdrawal, please see our Privacy Policy .

Reviewer #1: No

Reviewer #2: No

---

## [Author Response · Author response to Decision Letter 1]

13 Nov 2024

AE comments:

Specific Comments

Abstract. Include main research questions/hypotheses.

L42. Change greenspace to green space here and throughout the ms.

Corrected

L46. One point that has come up that may want to note is that redlining was not the singular factor that has been recently described, but one of several that may co-occur, etc. You mostly note this point already, but may touch upon a bit more that it isn’t just a simple classification that explains the findings of other studies. Likewise, this point also relates to the reviewer’s comment about zoning.

We have added a section focused on how zoning also plays a role in determining both the trend of homeownership and demographics, as well as the potential for HOLC C and D areas to be prime candidates for gentrification. However, as we explain in this section and in the associated figures, unfortunately the zoning data that is available is not usable to create a legitimate time-series. Additionally, we provide our reasons for using HOLC categories due to the fact they offer a standardized description of the various categories prior to subsequent lending impacts, and their fixed spatial extents provide a consistent location to analyze change over time. We do feel that by adding this section and highlighting that there is a greater flexibility in land use regulations in redlined districts, and that redlined districts are more vulnerable to shifts in zoning because they have fewer areas zoned single-family residential better make the case the HOLC boundaries were not a singular factor in driving land use and demographic trends, but are strongly linked to other regulatory processes.

L101-129. Much of this text can be moved to Methods or deleted. L101 should start the final paragraph that lays out the overarching goal/question followed by the specific questions/hypotheses/objectives.

We have incorporated this section into the Methods sub-section “Analysis of land-use diversity”, and removed any redundant text.

L130. I would suggest revising this first sentence a bit to indicate the gap in knowledge that you are seeking to answer.

The paragraph header has been adjusted to bring the main gaps in knowledge (the lack of time-series data, the importance of considering both land use and demographics) to the forefront before introducing our research questions.

L174. Please note here or elsewhere in the Methods what datum and projection were used in your geographic analyses.

Corrected

L192. How are you dealing with different resolutions of the datasets?

The resolution of the Drummond and Chen data sets are the same (1 m). We have edited the text to ensure that point is clear.

L215. Revise to ‘This approach…’

Corrected

L223. Did you evaluate collinearity amongst terms first? If not, please do. Also, did you have to transform any of the data? Finally, are you just using p-values here or AICc?

We have added text explaining how multicollinearity was expected between factors of race, education, and income. So we all demographic models were done independent from each other to isolate the specific demographic patterns.

L241. Here or elsewhere in Methods indicate the statistical software you used for analyses.

Corrected

L244. I would suggest cutting this sentence. Results here should be very succinct. You found x relationship, with y parameters being important.

Corrected

Fig. 2 legend. Delete ‘We determined significance through a generalized linear model with interactions between HOLC category and years.’ Also, no description of panel B is noted.

Corrected.

L276. Can revise to just indicate result, do not need to describe the statistical approach used as that was presented in the Methods.

We have removed superfluous text which was already described in the Methods section

L277. Provide the full statistical results (F = xxx, DF = x,y, p = z.zz) here and elsewhere in Results. Also, delete the sentence that starts at this line as you already explained post hoc approach in Methods.

Corrected and appreciated! On producing the statistics the authors discovered a typo in the statistical code, when corrected we found that HOLC D was significantly different in Shannon Diversity back to 1937. This actually bolsters our argument that redlining practices were not the creative force for present day land use segregation, but did correlate with increases in segregation of land use diversity in the following decades. We have corrected this text here and in the discussion.

L290. See previous comment on including detailed stats.

L301, L315, L358. Provide the statistical details of the analyses here.

We have added statistical details for all linearly modeled relationships to the text

Fig. 3 legend. Delete ‘We determined significance through a generalized linear model with interactions between HOLC category and years.’

Corrected

Fig. 4 legend. Delete ‘We determined significance through a generalized linear model with interactions between HOLC category, income class, and years.’

Corrected

L354. Move this item to Methods.

We have moved to the methods and added text on how we conducted the analysis.

Fig. 5 legend. Delete ‘We determined significance through a generalized linear model with interactions between HOLC category and years.’ And ‘Regression line equations and p values by HOLC category are provided.’

Corrected

L374. Can cut most of first two sentences. Begin with answers to your questions.

Corrected

Reviewer 1 Comments:

Reviewer #1: The principal claim of this article is succinctly summarized in the title: “Land use and socioeconomic time-series reveal legacy of redlining on present-day gentrification within a growing United States city,” but the study asks and answers much more far-reaching claims than this. And, the study effectively qualifies the specific role of “redlining,” meaning the impact of the 1938 HOLC Security Maps as a causal factor in this overall process. Instead, the authors use the HOLC categories as a snapshot of the land use and demographic conditions at the starting-point in their longitudinal study. The goal at the heart of the study is to track long-term changes in land use diversity and demographic areal composition from 1937 to 2019. While the long-term correlations between HOLC categories (developed originally and used by lending agencies thereafter for the purpose of qualifying applicants for mortgages) and downstream social inequality, urban amenities, environmental injustice, and other factors has been well-documented now in numerous studies amply cited in this study, the key innovation here is the use of an allegedly unique—for Denver-- high-resolution land-use data series (Drummond et al 2019) , for Denver in the years 1937, 1957, 1977, and 1997. This data series allows the authors to a) set a benchmark of land use prior to any possible impact of the HOLC maps, and b) track changes in 20-year intervals between the onset of the redlining era and pour contemporary, particularly the first two decades of the 21st century.

In the authors’ own words, “The spatial patterns of this connection between the built environment, past discriminatory policies, and the potential vulnerability to neighborhood gentrification have been connected from the present to the HOLC boundaries (30), but to understand how these spatial patterns emerged over time, time series data is needed. (96-101). The authors address three specific questions: “1. How have patterns in land use and demographics developed and diverged in formally categorized HOLC areas across Denver from 1937 to 2019? 2. How has the diversity of demography in Denver neighborhoods connected to HOLC areas changed over the same period? 3. How have patterns of inequality in land use and demographics from the previous decades influenced rapid present-day changes in Denver’s demographics and resulting neighborhood-specific patterns in gentrification?” (129-138).

Overall the authors are successful in answering all three of these questions, but there is a major factor that they both recognized and fail to include in their analyses, which calls into question the whole rationale for depending on HOLC classifications in the first place: Denver’s zoning ordinances.

The authors do show statistically significant patterns in which the land-use diversity was reinforced in HOLC category D areas across the entire time period, which residential, single-family land use also increased in areas A and B. Proportions of college-educated, white, and higher-income households were all significantly lower for HOLD D areas than for HOLC-A areas over time. Given the goals of the study, a significant achievement is that the authors are able to show significant changes within the overall trends from 1937-2019 in two time-periods: 1) a significant impact of urban renewal policies and projects during the 1960-1980 interval, and 2) the impact of gentrification in areas previously disadvantaged by the longer-term trends, in the early 20th century period.

While all of this is very solid statistically, this reviewer believes that the authors made a critical methodological mistake by not identifying and including the geographic boundaries of the city’s zoning ordinances, along with the land-use restrictions codified by those ordinances for those geographic boundaries, as independent variables in addition of the HOLC boundaries. The HOLC boundaries, after all, were established for the single purpose of evaluating the loan-security risk of residential properties. Thereafter, they were used or adapted by banks for decades, but only for the purpose of home loan qualifications. The principal innovation of this study is to use the Drummond, et al land-use maps, which capture all categories of land use. The authors use these maps primarily along the dimension of “diversity,” which is essentially bi-variate in their usage: they seek to distinguish high-diversity areas (where industrial and other non-residential uses are concentrated), and low-diversity, where primarily residential areas are concentrated. Zoning ordinances were invented in the early 20th century to exclude industrial and other commercial uses from certain residential areas. (and also to optimize industrial development). They did not exclude residential development within industrial/commercial areas, but they did exclude industrial/commercial development in areas newly developed to be exclusively residential. This all happened concurrent with the rise of racial restrictions, usually to exclude non-whites from those same all-residentially zoned areas. While this reviewer is not familiar with Denver’s specific zoning ordinances, based on their use in cities with which the reviewer does have familiarity, it is nearly certain that they were used to exclude land use “diversity” from residential areas over this long period. It is also likely that they were revised significantly over the years. It is also certain that zoning ordinances have a direct impact on land use, while HOLC categories have only an indirect impact. And, it is very well known that residential property values are directly impacted by proximity to nuisances typical of industrial and commercial land uses. It is likely, therefore, that the exclusivity of HOLC areas A and B, and the land use diversity of areas C and D, can be explained entirely by the regulatory effect of these ordinances. The word “zoning” only appears three times in this manuscript (95, 403, 416), but in each of these passages the authors acknowledge the importance of zoning ordinances.

The authors laudably avoid assigning HOLC maps a singular causal role in the longer temporal patterns of land use and demographic spatial inequality. They show empirically that the inequalities of land use and demographic geography pre-existed the HOLC maps, and that changes occurred long after their implementation. This reviewer believes that the failure to consider the role of zoning geographies across this period is a major oversight. While the HOLC maps have come to stand for a long-term inscription of inequality in urban landscapes, the authors needn’t have tied their study so closely to those maps which, as they acknowledge, do not even cover downtown areas. The status quo of land use and social demography in 1937 and subsequent years could have been classified by an independent set of area units, and tracked over time for long-term patterns. The use of zoning geographies would have gone a long way toward helping to explain those patterns, as city planning authorities were required to certify all building project according to these maps—and not to the HOLC maps. This reviewer does not recommend removing the HOLC geographies from the study—far from it: the inclusion of the HOLC geographies and categories should still be an important set of variables in the study. But they should be used in conjunction with the explanatory power of the zoning geographies. This approach will a) give a much more accurate view of the long-term urban developmental historical process (recognizing the factor most likely to shape land use), and b) help to advance the latest state of the art in understanding the impact of the HOLC maps. My recommendation, therefore, does not imply scrapping the entire approach of this study. Instead, it suggests the addition of mapping layers with associated attributes, most probably chosen to coincide with the four land-use data points of 1937, 57, 77, and 97, and re-running their analyses to include the independent variable of land use restrictions on every space in the study.

We greatly thank Reviewer 1 for their in-depth consideration of this work. We also completely agree with them! Previously we had not included any analysis on zoning (aside from the comments we had already made expressing that zoning ordination would also be considered a driving force behind patterns in land use configuration and diversity. This was borne out of a limited access to historical zoning data. However, thanks to this comment we have engaged with the GIS office of the City of Denver and received some historical zoning data. However, this data only goes back to 1955, and only indicated the year when a zoning ordinance was changed, but it does not indicate exactly what the previous ordinance was. Thus, we cannot exactly track a similar time-series of zoning influence on land use as we do with the rest of the analysis.

However, we have added a section focused on how zoning also plays a role in determining both the trend of homeownership and demographics, as well as the potential for HOLC C and D areas to be prime candidates for gentrification. However, as we explain in this section and in the associated figures, unfortunately the zoning data that is available is not usable to create a legitimate time-series. Additionally, we provide our reasons for using HOLC categories due to the fact they offer a standardized description of the various categories prior to subsequent lending impacts, and their fixed spatial extents provide a consistent location to analyze change over time.

With the limited zoning data we have we were able to present some analysis that reflects similar patterns to what we have shown in regards to land cover change. Firstly, we have found that former HOLC D areas currently are zoned to include a greater diversity of zoning ordinances, and moreover, by tracking the dates of zoning change, we found HOLC C and D areas experienced more regular changes in their zoning ordinances over a period of 1955-2020. These results are now included as Supplemental Figure 2.

We do feel that by adding this addition text, analysis, and highlighting that there is a greater flexibility in land use regulations in redlined districts, and that redlined districts are more vulnerable to shifts in zoning because they have fewer areas zoned single-family residential better make the case the HOLC boundaries were not a singular factor in driving land use and demographic trends, but are strongly linked to other regulatory processes.

Reviewer 2 Comments:

Reviewer #2: This research analyzed changes in land use and socioeconomic factors, considering the impacts of historical redlining, to identify current patterns of gentrification in Denver, USA. They used boxplots and statistical a

---

## [Decision Letter · Decision Letter 1]

9 Jan 2025

Land use and socioeconomic time-series reveal legacy of redlining on present-day gentrification within a growing United States city.

PONE-D-24-05471R1

Dear Dr. Ibsen,

We’re pleased to inform you that your manuscript has been judged scientifically suitable for publication and will be formally accepted for publication once it meets all outstanding technical requirements. Both reviewers and I appreciate all the revisions to the text and believe this will be a valuable addition to the literature.

Kind regards,

Christopher A. Lepczyk

Academic Editor

PLOS ONE

Additional Editor Comments (optional):

Reviewers' comments:

Reviewer's Responses to Questions

**Comments to the Author**

1. If the authors have adequately addressed your comments raised in a previous round of review and you feel that this manuscript is now acceptable for publication, you may indicate that here to bypass the “Comments to the Author” section, enter your conflict of interest statement in the “Confidential to Editor” section, and submit your "Accept" recommendation.

Reviewer #1: All comments have been addressed

Reviewer #2: All comments have been addressed

2. Is the manuscript technically sound, and do the data support the conclusions?

Reviewer #1: Yes

Reviewer #2: Yes

3. Has the statistical analysis been performed appropriately and rigorously? 

Reviewer #1: Yes

Reviewer #2: Yes

4. Have the authors made all data underlying the findings in their manuscript fully available?

Reviewer #1: Yes

Reviewer #2: Yes

5. Is the manuscript presented in an intelligible fashion and written in standard English?

Reviewer #1: Yes

Reviewer #2: Yes

6. Review Comments to the Author

Reviewer #1: I am very happy with the authors´ response to my recommendation that they include historical Denver zoning data in their analysis. They have incorporated a fresh analysis of the zoning data in a convincing way that reinforces their previous analysis and conclusions. Note a very tiny detail, a typo on line 805 describing supplemental figure 2, reads "differences," should be "different."

Reviewer #2: Thank you to the authors for their efforts in addressing my comments and suggestions. I believe the revised version has reached a publishable level.

However, I have a few minor points to mention, which I believe can be addressed during the galley proof:

1. In Fig. 2, the sub-label (a) is missing in the current version.

2. Some paragraphs, such as those on pages 24 and 25, are too long. I recommend dividing these paragraphs to enhance readability.

7. PLOS authors have the option to publish the peer review history of their article (what does this mean? ). If published, this will include your full peer review and any attached files.

**Do you want your identity to be public for this peer review?** For information about this choice, including consent withdrawal, please see our Privacy Policy .

Reviewer #1: No

Reviewer #2: No

---

## [Editor Report · Acceptance letter]

PONE-D-24-05471R1

PLOS ONE

Dear Dr. Ibsen,

I'm pleased to inform you that your manuscript has been deemed suitable for publication in PLOS ONE. Congratulations! Your manuscript is now being handed over to our production team.

Kind regards,

on behalf of

Dr. Christopher A. Lepczyk

Academic Editor

PLOS ONE